# Fucoxanthin, A Carotenoid Derived from *Phaeodactylum tricornutum* Exerts Antiproliferative and Antioxidant Activities In Vitro

**DOI:** 10.3390/antiox8060183

**Published:** 2019-06-19

**Authors:** Ulrike Neumann, Felix Derwenskus, Verena Flaiz Flister, Ulrike Schmid-Staiger, Thomas Hirth, Stephan C. Bischoff

**Affiliations:** 1Institute of Clinical Nutrition, University of Hohenheim, Fruwirthstr. 12, 70593 Stuttgart, Germany; ulrike.neumann@uni-hohenheim.de (U.N.); verena.flister@gmail.com (V.F.F.); 2Fraunhofer Institute for Interfacial Engineering and Biotechnology IGB, Nobelstr. 12, 70569 Stuttgart, Germany; felix.derwenskus@igb.fraunhofer.de (F.D.); ulrike.schmid-staiger@igb.fraunhofer.de (U.S.-S.); 3Institute of Interfacial Process Engineering and Plasma Technology IGVP, University of Stuttgart, Nobelstr. 12, 70569 Stuttgart, Germany; 4Karlsruhe Institute for Technology, Kaiserstr. 12, 76131 Karlsruhe, Germany; thomas.hirth@kit.edu

**Keywords:** microalgae, *Phaeodactylum tricornutum*, fucoxanthin, antioxidative, antiproliferative

## Abstract

Microalgae contain a multitude of nutrients and can be grown sustainably. Fucoxanthin, a carotenoid from *Phaeodactylum tricornutum,* could have beneficial health effects. Therefore, we investigated the anti-inflammatory, antioxidative and antiproliferative effects of fucoxanthin derived from this diatom in vitro. The effects of purified fucoxanthin on metabolic activity were assessed in blood mononuclear cells and different cell lines. In cell lines, caspase 3/7 activity was also analyzed. Nitrogen monoxide release and mRNA-expression of proinflammatory cytokines were measured. For antioxidant assays, cell free assays were conducted. Additionally, the antioxidant effect in neutrophils was quantified and glutathione was determined in HeLa cells. The results show that neither did fucoxanthin have anti-inflammatory properties nor did it exert cytotoxic effects on mononuclear cells. However, the metabolic activity of cell lines was decreased up to 58% and fucoxanthin increased the caspase 3/7 activity up to 4.6-fold. Additionally, dose-dependent antioxidant effects were detected, resulting in a 63% decrease in chemiluminescence in blood neutrophils and a 3.3-fold increase in the ratio of reduced to oxidized glutathione. Our studies show that fucoxanthin possesses antiproliferative and antioxidant activities in vitro. Hence, this carotenoid or the whole microalgae *P. tricornutum* could be considered as a food or nutraceutical in human nutrition, showcasing beneficial health effects.

## 1. Introduction

Microalgae are microscopic small unicellular organisms that are abundant in various habitats around the globe. They can be cultured in open ponds or photobioreactors without the use of arable land and thus represent a promising sustainable alternative to plant-based proteins. They do not only produce high amounts of proteins but are also a good source for fatty acids, and vitamins, which can provide health promoting effects in a human diet. Hence, they can serve as novel functional foods or be incorporated into existing food products [1,2].

The unicellular pennate diatom *Phaeodactylum tricornutum* exists in three different morphotypes and its genome has been already fully sequenced [3]. The diatom contains a multitude of different components that could provide health beneficial effects. These include the omega-3 fatty acid eicosapentaenoic acid (EPA), polyphenols like (epi-) catechin and oxygenated carotenoids like fucoxanthin [4,5,6,7,8]. Fucoxanthin, a major marine carotenoid, is located in the thylakoids of chloroplasts and forms a light harvesting complex (LHC) with chlorophyll a/c [9,10]. It was shown in several studies, that fucoxanthin from macroalgae possesses health-conducive effects, including anti-inflammatory, antioxidant, antiobesity and anticancer activities [11,12,13,14,15]. Since most of these studies used fucoxanthin derived from macroalgae, only little is known about the health-beneficial effects of fucoxanthin derived from *P. tricornutum*, although this microalga was shown to have a high content of fucoxanthin between 16.5 to 26.1 mg per gram dry matter (dm) [16,17]. Thus, diatom biomass can contain up to ten times more fucoxanthin than macroalgae. Additionally, diatoms, different from macroalgae, can be cultivated indoors and outdoors not only during specific seasons but all around the year with a high biomass productivity [18]. It is already demonstrated that they can be produced in various types of different closed photobioreactors, like bubble columns and flat panel airlift bioreactors [19,20,21]. Hence, *P. tricornutum* might be a suitable source for the commercial production of fucoxanthin.

The prevalence of diseases linked to oxidative stress and inflammation is constantly increasing in developed countries [22,23,24]. The occurrence of those diseases can also be linked to the emergence of cancer [25,26]. Therefore, research is trying to identify new compounds that could be used to reduce inflammation, oxidative stress, and the viability of cancer cells. Fucoxanthin derived from *P. tricornutum* might be used as a new nutraceutical, if it also exhibits the health beneficial effects that were already shown for the carotenoid derived from macroalgae. 

## 2. Materials and Methods 

### 2.1. Cultivation of Microalgae and Fucoxanthin Extraction

*P. tricornutum* UTEX 640 was cultivated in 180 L Flat-Panel-Airlift photobioreactors in an outdoor pilot scale plant located at the Fraunhofer Center for Chemical-Biotechnological Processes CBP in Leuna, Germany. Modified Mann and Myers medium was used as culture medium as described in Meiser et al. [19,27]. The biomass was disrupted using stirred ball milling (PML-2, Bühler) and freeze-drying prior to fucoxanthin extraction. The cell disruption and extraction method were previously described in detail in Derwenskus et al. [17]. Briefly, Fucoxanthin was extracted from the disrupted biomass by pressurized liquid extraction (ASE 350, Thermo-Fisher) for a static extraction time of 20 min at 100 °C and 100 bar using adequate subcritical organic solvents (described in [28]). Subsequently, the fucoxanthin was purified by multiple separation steps using filters (0.25 µm) consisting of polytetrafluoroethylene to a final purity of 99.2% (*w/w*) (HPLC). It was compared to a commercial fucoxanthin standard (16337, Sigma-Aldrich) using UHPLC-MS.

It was compared to a commercial fucoxanthin standard (16337, Sigma-Aldrich) using UHPLC-MS.

### 2.2. Determination of Fucoxanthin by HPLC and UHPLC-MS

Fucoxanthin was quantified using the HPLC method described by Gille et al. [28] with slight modifications. Briefly, the purified fucoxanthin was resolved in pure ethanol with BHT (250 mg/L) and compared to a commercial analytical standard (16337, Sigma-Aldrich, St. Louis, MO, USA) using reverse-phase HPLC with a Suplex pKb 100 column (5 µm, 250 × 4.6mm, Supelco, Bellefonte, PA, USA). Samples (injection volume 5 µL) were analyzed using a HPLC (1200 Infinity, Agilent, Santa Clara, CA, USA) equipped with a multi-wavelength UV detector at 450 nm and a flow rate of 1 mL/min. The gradient used for the method is described in detail elsewhere [29]. 

Additionally, fucoxanthin from *P. tricornutum* was analyzed and compared to the commercial standard by UHPLC-DAD (1290 Infinity, Agilent Technologies) using a Zorbax Eclipse Plus C18 (2.1 × 50 mm) column with a particle size of 1.8 µm. Mobile phase A contained water with 0.1% formic acid and mobile phase B consisted of methanol with 0.1% formic acid. The gradient used is shown in Table 1. The fucoxanthin was detected at 450 nm and analyzed in a mass spectrometer (LTQ XL, Thermo Scientific) using ESI in full scan mode from 200 to 1000 *m/z* at a temperature of 275 °C and −9.0 V. The *m/z* values (see Appendix A) were compared to the analytical standard and to literature [30]. 

### 2.3. Isolation of Human Primary Blood Cells Band Cell Cultures

Anticoagulated blood was collected from healthy volunteers, approved by the local ethics committee (F-2015-064, Landesärztekammer Baden-Württemberg). Isolation of polymorphonuclear leukocytes (PML) and peripheral blood mononuclear cells (PBMCs) via dextran sedimentation and density gradient centrifugation was conducted as previously described by El Benna & Dang (2007) [30] and Neumann et al. (2018) [31]. PMLs were resuspended in DPBS, PBMCs in RPMI medium with 10% fetal calf serum and 1% penicillin / streptomycin. HepG2 cells were provided by the Max Rubner-Institute (Karlsruhe, Germany). RAW 264.7, HepG2, Caco-2 and HeLa cells were cultured in DMEM with 10% fetal calf serum and 1% penicillin / streptomycin. 

### 2.4. Metabolic Activity

Metabolic activity was assessed using the tetrazolium dye 3-(4,5-dimethylthiazol-2-yl)-2,5-diphenyltetrazolium bromide (MTT). PBMCs (3 × 10^5^ cells per well) and RAW 264.7 cells (1 × 10^4^ cells per well) were incubated with fucoxanthin (0.1, 1, 10 and 50 µg/mL), β-carotene (1 µg/mL), dimethyl sulfoxide (DMSO, 0.1%) as solvent control or DMSO (5%) as positive control for 24 h. Subsequent, MTT assay was conducted as previously described [31]. 

### 2.5. Antiinflammatory Assays

Nitric oxide (NO) production in RAW 264.7 cells was measured using the Griess assay. Therefore, 5 × 105 cells were treated with fucoxanthin (0.1 and 1 µg/mL) or β-carotene (1 µg/mL) and stimulated with lipopolysaccharide (LPS, 1 µg/mL) for 24 h. The level of NO was measured using the Griess reaction, as previously described [32].

The effects of fucoxanthin on the LPS-induced mRNA expression of the cytokines interleukin- (IL-)1β, IL-6, tumor necrosis factor α (TNF-α) and the enzyme cyclooxygenase-2 (COX-2) in PBMCs was measured by quantitative real-time polymerase chain reaction (qRT-PCR). Cells were incubated with fucoxanthin (0.1, 1 or 10 µg/mL), β-carotene (1 µg/mL) or DMSO (0.1%) as solvent control for 24 h. Cells were stimulated with LPS (1 µg/mL) for 6 h. The mRNA expression was measured as formerly described [31].

### 2.6. Antioxidant Assays

Total phenolics content (TPC) was determined using the Folin-Ciocalteu method [33] with minor modifications. In a 96-well microplate, 30 µL of fucoxanthin (0.1, 1 or 10 µg/mL) were mixed with 150 µL Folin-Ciocalteu reagent (diluted 1/10 in water) and 120 µL sodium carbonate solution (75 g/L). To obtain an individual blank, samples were mixed with 120 µL sodium carbonate solution and 150 µL water. After 2 h in the dark at room temperature, the absorbance was measured at 765 nm with a BioTek Synergy HT plate reader (BioTek Instruments, Winooski, VT, USA). Gallic acid was used to establish a calibration curve (30–580 µM) and results are expressed as gallic acid equivalents (GAE) per gram dry matter (dm).

The ferric reducing antioxidant power (FRAP) assay was performed in accordance to the method of Benzie and Strain [34]. An individual blank was measured for each sample. Ferrous sulphate solutions were used for calibration (50–1000 µM) and the results are expressed as mmol Fe^2+^ per gram dm.

For the 2,2-diphenyl-1-picrylhydrazyl (DPPH) assay a calibration curve was obtained by using DPPH concentrations in the range of 0-100 µM. A DPPH solution was freshly prepared in methanol. 150 µL DPPH (0.1 MM) were mixed with 50 µL fucoxanthin (0.1, 1, 10 or 50 µg/mL), 150 µL methanol instead of DPPH were used for blank measurement. The percentage inhibition was calculated using the following formula:%Inhibition = (OD_DPPH_ − OD_Sample_)/OD_DPPH_ × 100

The half maximal inhibitory concentration (IC50) was calculated by linear regression, plotting the percentage inhibition against the different extract concentrations.

The glutathione (GSH) to glutathione disulfide (GSSG) ratio as a marker for oxidative stress was determined using the GSH/GSSG-Glo™ Assay (Promega, Mannheim, Germany). 2 × 104 HeLa cells were incubated with fucoxanthin (0.1, 1, 10 and 50 µg/mL), β-carotene (1 µg/mL), DMSO (0.1%) as solvent control or menadione (40 µM) as control for 24 h.

2′,7′-dichlorofluorescin (DCF) fluorescence and ROS production using luminol chemiluminescence were measured in human PMLs. For DCF fluorescence 100 µL of freshly drawn blood were incubated with 2′,7′-dichlorofluorescin diacetate (20 µM) at 37 °C for 15 min. Lipopolysaccharide (10 ng/mL) and fucoxanthin (0.1, 1, 10 or 50 µg/mL) or β-carotene (1 µg/mL) were added for 1 h at 37 °C. Cells were then stimulated with N-formylmethionyl-leucyl-phenylalanine (fMLP, 500nM). The reaction was stopped after 5 min by placing the samples on ice. After red blood cell lysis using the BD FACS lysing solution according to the manufacturer’s instructions, DCF fluorescence was measured with a BD FACS Canto II (BD Biosciences, Becton, Dickinson and Company, San Jose, CA, USA). The percentage of fluorescent PMLs was calculated using the BD FACS Diva Software.

In luminol assays, PMLs were incubated with fucoxanthin (0.1, 1, 10 and 50 µg/mL), β-carotene (1 µg/mL), DMSO (0.1%) as solvent control or menadione (40 µM) as control. The assay was conducted as previously described [35] and cells were stimulated with phorbol 12-myristate 13-acetate (PMA, 100 ng/mL). Chemiluminescence was measured in a Berthold-Biolumat LB937 (Berthold Technologies Co., Bad Wildbad, Germany) at 37 °C for 15 min. Percentage inhibition of luminescence was calculated using the area under the curve (AUC) values.

### 2.7. Cytotoxic and Apoptotic Assays

MTT assays were conducted with Caco-2, HeLa and Hep G2 cells to assess cytotoxic activity of fucoxanthin on cancer cells. Cells were incubated with fucoxanthin (0.1, 1, 10 and 50 µg/mL), β-carotene (1 µg/mL), DMSO (0.1%) or staurosporine (1 µM) for 48 h, as described elsewhere [32]. For apoptotic properties, the caspase-glo® 3/7 assay (Promega, Mannheim, Germany) was conducted according to the manufacturer’s instructions.

### 2.8. Statistics

Data are expressed as mean ± standard error of the mean (SEM) in graphs or ± standard deviation (SD) in tables. Graphs were generated using GraphPad Prism 5 (La Jolla, CA, USA), statistics were done using IBM SPSS Statistics 25 (IBM Corp., Armonk, NY, USA), graphs. Normal distribution was tested using the Shapiro–Wilk test. One-way analysis of variance (ANOVA) was used to evaluate statistic significant differences (*p* < 0.05) between groups. The equality of variances was tested utilizing Levene’s test. For equal variances, Tukey’s HSD post hoc test was used; for unequal variances, Dunnett’s T3 post hoc test3. 

## 3. Results

### 3.1. Metabolic Activity

For PBMCs, only the positive control with a final concentration of 5% DMSO led to a significant decrease in metabolic activity analyzed using the MTT assay. Fucoxanthin up to a concentration of 50 µg/ml did not influence the metabolic activity (Table 2). For RAW 264.7, however, a decrease in metabolic activity was shown for the positive control and the highest fucoxanthin concentration tested (50 µg/mL, Table 2). Hence, the following anti-inflammatory assays were only conducted with non-cytotoxic concentrations.

### 3.2. Antiinflammatory Effects

To analyze the anti-inflammatory effects of fucoxanthin, the NO production of RAW 264.7 cells was measured, and mRNA-expressions of inflammatory cytokines were determined in PBMCs (Table 3). Neither fucoxanthin nor β-carotene showed anti-inflammatory effects in the tested concentrations (Table 3). Additionally, no pro-inflammatory effect was seen in unstimulated cells (Table 3).

### 3.3. Antioxidant Effects

Various assays were conducted to analyze the antioxidant properties of fucoxanthin. In the DPPH assay, fucoxanthin had an IC50 value of 201.2 ± 21.4 µg/mL, while the value for ascorbic acid was 70.3 ± 18.7 µg/mL and for astaxanthin 79.32 ± 18.10 µg/mL. Due to low antioxidant effects, an IC50 for β-carotene could not be calculated. Results of the FRAP assay show that fucoxanthin is equivalent to 64.74 ± 3.93 mmol Fe^2+^ per gram dm, β-carotene to 6.55 ± 0.33 and astaxanthin to 63.97 ± 6.79 mmol Fe^2+^ per gram. 

Neither fucoxanthin nor β-carotene showed effects on the DCF fluorescence of PMLs (data not shown). Menadione decreased the GSH/GSSG ratio. However, according to statistical analysis this change was not significant (*p* = 0.09). Fucoxanthin in the highest concentration (50 µg/mL) increased the ratio significantly (Figure 1A). Lower concentrations, however, showed no effects. Luminol chemiluminescence was measured to evaluate the antioxidant properties of fucoxanthin in PMLs. The results are depicted in Figure 1B and show that menadione decreased the luminescence by 96%, fucoxanthin by 63% at 50 µg/mL. The antioxidant effect is dose-response dependent.

### 3.4. Cytotoxic and Apoptotic Effects

Fucoxanthin was able to reduce the metabolic activity of Hep G2, HeLa and Caco-2 cells in a dose-dependent manner (Figure 2A–C). An inhibitory effect of up to 58% was measured in Hep G2 cells. In HeLa and Caco-2 cells, the effect was stronger than that of the positive control with a final concentration of 5% DMSO. In order to evaluate if the decrease in metabolic activity is linked to apoptosis, the caspase 3/7 activity was determined. The results show that fucoxanthin led to a dose-dependent increase in caspase 3/7 activity (Figure 2D–F). A 4.6-fold increase in caspase activity was measured in HeLa cells for the highest fucoxanthin concentration. 50 µg/mL of the carotenoid led to a higher caspase 3/7 activity than 1µM staurosporine in all tested cell cultures.

## 4. Discussion

Fucoxanthin, a major marine carotenoid that is up to now obtained from macroalgae, was successfully extracted from the microalgae *P. tricornutum*. In this study, we found that fucoxanthin had no influence on the metabolic activity of PBMCs, which was also reported by Ishikawa et al. [36]. This leads to the assumption that fucoxanthin, up to concentrations of 50 µg/mL, has no cytotoxic effects on these cells. However, the carotenoid had a cytotoxic effect on the mouse macrophage cell line RAW 264.7 at 50 µg/mL. This is not concordant to a study by Islam et al. (2013), who only reported a reduced cell viability at much higher doses of fucoxanthin [37]. 

This study found no effects of fucoxanthin on the NO production of LPS-stimulated RAW 264.7 cells. A study by Islam et al. (2013) supports these findings [37]. Authors showed anti-inflammatory effects only at higher concentrations [37]. However, other studies were able to show a dose-dependent inhibition of NO production by fucoxanthin at lower concentrations [12,38,39]. Here, we also reported that the carotenoid had no effect on the mRNA-expression of pro-inflammatory cytokines in human PBMCs. This is also not concordant to the study by Heo et al. (2010), who showed an inhibitory effect on the mRNA expression in RAW 264.7 cells [12]. Yet, to date, no studies on the anti-inflammatory effects of fucoxanthin on human primary blood cells have been published. It is noteworthy that all mentioned studies utilized fucoxanthin extracted from seaweeds. Therefore, it can be assumed that the carotenoid from *P. tricornutum* might have different effects or that the measured effects in other studies are based on impurities resulting from extraction.

Fucoxanthin showed strong antioxidant effects in cell-free and cell-based assays. The IC50 concentration of fucoxanthin with 201 µg/mL in the DPPH assay was higher than that reported by Sachindra et al. (2007) [40]. The higher values might be caused by differences in extraction or on the origin of the carotenoid. As previously described, β-carotene showed no DPPH radical scavenging activity [41]. Antioxidant effects of ascorbic acid, however, were stronger in FRAP and DPPH assays. The FRAP assay revealed that the antioxidant effects of fucoxanthin extracted from P. tricornutum does not significantly differ from that of astaxanthin, a carotenoid with strong antioxidant effects derived from the red algae *Haematococcus pluvialis* which is already successfully commercialized [42,43]. Both, fucoxanthin as well as astaxanthin showed a strong antioxidant effect in the FRAP assay compared to β-carotene. 

In the DCF fluorescence assay, neither fucoxanthin nor β-carotene showed antioxidant effects while the luminol assay revealed dose-dependent antioxidant properties of fucoxanthin. The DCF assay is used for the intracellular detection of ROS, especially H2O2 [44]. Luminol, on the other hand, can detect the sum of extra- and intracellular ROS, especially those generated by the myeloperoxidase [45]. Hence, it can be assumed that fucoxanthin is either inhibiting the myeloperoxidase activation or quenching the bactericidal hypochlorite produced by this enzyme. Fucoxanthin was also able to increase the GSH level in HeLa cells, which was already shown by a study in human keratinocytes [46]. GSH as an antioxidant is able to scavenge ROS; the ratio of GSH to GSSG is often used as a marker for oxidative stress. A multitude of diseases is linked to a decreased GSH to GSSG ratio, including Alzheimer’s and Parkinson’s disease [47,48,49]. Fucoxanthin could help to increase reduced GSH and hence ameliorate the negative effects of oxidative stress.

Although fucoxanthin had no effect on the metabolic activity of human blood cells, a dose-dependent influence on different carcinoma cell lines was shown. For Caco-2 cells and different cell lines, this was already shown previously [50,51,52]. To analyze if the reduced metabolic activity is linked to an increased apoptosis of cells, we also measured the caspase 3/7 activity. An increase in activity was shown for all cells, leading to the conclusion that fucoxanthin from P. tricornutum is able to induce apoptosis in different cancer cells. Kim et al. (2010) were able to show, that the induced apoptosis is caused by the formation of ROS by fucoxanthin [51]. The authors state that the production of intracellular H_2_O_2_ and superoxide in the carcinoma cells triggers the apoptosis. This is, however, inconsistent with the antioxidant effects of the carotenoid that was shown in this study. On the other hand, Kotake-Nara et al. (2005) reported that the induced apoptosis is not accompanied by the production of ROS and caused by the loss of mitochondrial membrane potential [53]. It is assumed that oxidative stress is linked to the initiation and promotion of cancer [25]. ROS might lead to DNA damages and hence lead to uncontrolled cell proliferation and decreased apoptosis in cancer cells. On the other hand, antitumor drugs often function by producing ROS that induce oxidative stress in tumor cells and lead to cell death [54]. The role of antioxidants in tumor therapy is therefore a controversial issue. Some studies show that antioxidants can promote the outcome of therapy, while others show negative effects [54,55]. 

In summary, we were able to show antioxidant and antiproliferative but no anti-inflammatory effects of the carotenoid fucoxanthin extracted from the microalgae *P. tricornutum*. Fucoxanthin was able to inhibit the oxidative burst in human PMLs, scavenge radicals and increase the GSH to GSSG ratio. Additionally, the metabolic activity was decreased, and apoptosis increased by the carotenoid. This leads to the conclusion that fucoxanthin or the whole microalgae biomass, including fucoxanthin in high amounts, could be considered in nutrition in order to ameliorate the effects of diseases linked to oxidative stress. Additionally, fucoxanthin could help to support traditional cancer treatment because of its beneficial health effects. Human trials are needed in future to further support these suggestions. 

## Figures and Tables

**Figure 1 antioxidants-08-00183-f001:**
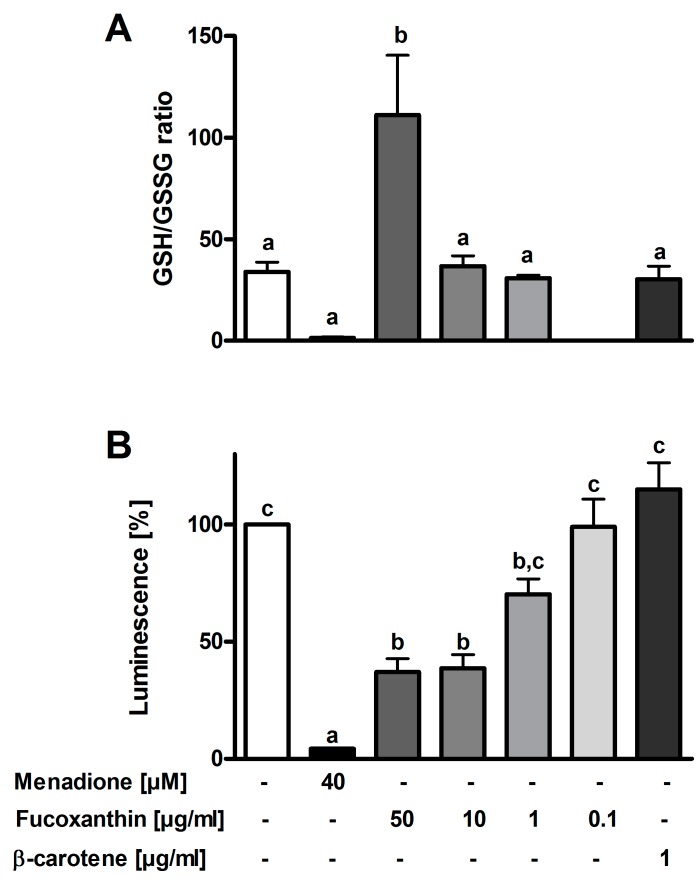
Effects of fucoxanthin from *P. tricornutum* on GSH to GSSG ratio in HeLa cells (**A**) and on luminol chemiluminescence in freshly isolated PMLs (**B**) (*n* = 3–5). Different letters represent significantly different groups (ANOVA followed by Tukey post hoc test for GSH/GSSG assay or with Dunnett’s T3 post hoc test for luminol chemiluminescence, *p* < 0.05). Abbreviations: GSH glutathione, GSSG glutathione disulfide.

**Figure 2 antioxidants-08-00183-f002:**
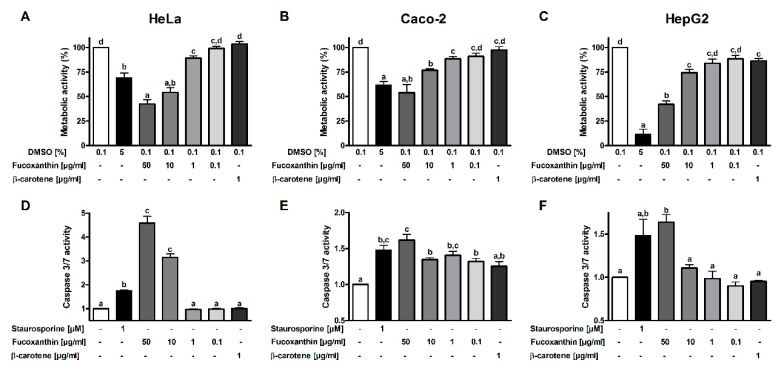
Effects of fucoxanthin from *P. tricornutum* on metabolic activity of HeLa (**A**), Caco-2 (**B**) and HepG2 (**C**) cells (*n* = 4–6). Cells were incubated for 48 hours, DMSO (5%) was used as a positive control. Caspase 3/7 activity as a marker for apoptosis was assessed in HeLa (**D**), Caco-2 (**E**) and HepG2 (**F**) cells (*n* = 4). Here, staurosporine (1 µM) was used as a positive control. Different letters mark significant differences (ANOVA followed by Tukey post hoc test for D or with Dunnett’s T3 post hoc test, *p* < 0.05).

**Table 1 antioxidants-08-00183-t001:** Solvent gradient for the UHPLC-MS-Method used in this study. Mobile phase A consisted of water with 0.1% formic acid and mobile phase B was methanol with 0.1% formic acid.

Time [min]	Mobile Phase A [%]	Mobile Phase B [%]
0	70	30
8	3	97
11	3	97
11.1	70	30
14	70	30

**Table 2 antioxidants-08-00183-t002:** Effects of fucoxanthin from P. tricornutum, vehicle control (DMSO, 0.1%), positive control (DMSO, 5%) and β-carotene on metabolic activity of PBMCs and RAW 264.7 cells. Data are presented as mean ± SD (*n* = 4–6).

	[µg/mL]	Metabolic Activity [%]	
		PBMCs	RAW 264.7
Vehicle control		100	100
Positive control		54.27 ± 4.23 **	3.39 ± 3.02 ***
Fucoxanthin	50	104.7 ± 14.16	20.88 ± 6.33 **
	10	98.76 ± 5.38	73.28 ± 9.60
	1	95.25 ± 3.84	89.64 ± 13.76
	0.1	88.35 ± 14.13	93.12 ± 5.09
β-carotene	1	87.65 ± 13.47	80.30 ± 13.09

Asterisks mark significant differences to vehicle control as analyzed by ANOVA with Dunnett’s T3 post hoc test (*<0.05, ** <0.01, *** <0.001).

**Table 3 antioxidants-08-00183-t003:** Effects of fucoxanthin from P. tricornutum, vehicle control (DMSO, 0.1%) and β-carotene on relative mRNA expression of pro-inflammatory cytokines in PBMCs and on NO production in RAW 264.7 cells. Data are presented as mean ± SD (*n* = 5–7).

	[µg/mL]	IL-1β [%]	IL-6 [%]	TNFα [%]	NO [µM]
		PBMCs	PBMCs	PBMCs	RAW 264.7
Vehicle control		100	100	100	59.66 ± 7.59
Fucoxanthin	10	110.1 ± 26.55	99.15 ± 26.14	160.4 ± 68.13	56.53 ± 6.58
	1	97.65 ± 18.09	92.10 ± 22.18	133.3 ± 60.35	60.84 ± 7.65
	0.1	101.3 ± 18.29	122.5 ± 32.23	133.9 ± 55.50	59.21 ± 7.69
β-carotene	1	75.07 ± 44.36.	104.2 ± 57.37	158.7 ± 80.90	59.84 ± 7.10
Vehicle control, unstimulated		0.2 ± 0.1	0.1 ± 0.1	5.6 ± 5.7	0.0 ± 0.0
Fucoxanthin, unstimulated	10	0.3 ± 0.5	0.1 ± 0.04	5.1 ± 6.2	0.03 ± 0.1
β-carotene, unstimulated	1	0.2 ± 0.2	0.1 ± 0.2	3.8 ± 2.2	0.0 ± 0.0

No differences between stimulated groups and no differences between unstimulated groups were found by ANOVA (*p* < 0.05). Abbreviations: IL interleukin; TNF tumor necrosis factor, NO nitrogen monoxide.

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
