# Peer review of "Fucoxanthin, A Carotenoid Derived from Phaeodactylum tricornutum Exerts Antiproliferative and Antioxidant Activities In Vitro"

_antioxidants, 2019, doi:10.3390/antiox8060183_

Round 1

Reviewer 1 Report

The authors have set out to evaluate the effect of fucoxanthin (FXN) in human leukocytes, and RAW cells stimulated with LPS as an in vitro model of inflammation.  Measurements of metabolic activity (via MTT assay) and anti-inflammatory effects (gene expression of cytokines and NO production) were performed in RAW cells.  In polymorphonuclear leukocytes the antioxidant effects of FXN was assessed. The effects of FXN were measured in 3 cell lines.

There are several critical issues with this study.

1 MTT assays
The concentrations of FXN uses are fairly high and range from ~76 microM at 50 microgram/mL to the lowest concentration of ~152 nM at 0.1 microgram/mL.  Unsurprisingly there is a reduction in MTT activity at the higher doses in cell lines.  PBMC have a relative lesser reduction, but would probably be less active than cell lines.

The use of the MTT assay here is asserted to be as a measure of metabolic activity, but this assay can also be used to infer the general health of the cells and can explain why functional activity is reduced.  In RAW cells cytokine gene expression was measured after treatment with FXN (Table 3) at a concentration that caused a 25% reduction in MTT activity (Table 2, 10ug/mL).  Given the caspase activity observed in HepG2 and HeLa cells (Fig 2) it is likely that FXN at these concentrations is causing cell death possibly by apoptosis.  This may account for the increased expression of TNF and IL1b expression at the 10ug/mL concentration, as dying cells releasing cellular content promote an inflammatory response if controlled apoptosis is not in effect.

2 Inflammatory model
The stimulation of RAW cells with LPS in differing concentrations of FXN does not have a "no LPS stimulation" control.  This is necessary as the extract of FXN may activate cells of its own accord. Given the relatively lesser effect that FXN has on PBMC compared to RAW cells it is surprising that the authors did not use LPS-stimulated PBMC. By carefully collecting most of the supernatant after 24hr LPS stimulation for cytokine ELISAs, the PBMC could have been used for MTT assay.

3 Apoptosis
Figure 2 indicates that there may be degrees in the efficacy of the apoptosis-inducing effect of FXN.  The effect appears to be weaker in Caco-2 cells, and by comparing with HeLa and HepG2 cells the authors may better be able to dissect the mechanism by which FXN induces its effects.  This data is the most compelling but is not particularly novel given previous work in this area (PMID:28373414).  These assays are described as "proliferative assays" but could only be described as indicative of proliferation.  Proliferation is better measured by entry into S-phase (BrdU assay or thymidine incorporation) or cell fission (dye dilution in CFSE labelled cells) or inferring cell division by DNA content (propidium iodide staining).

4 Composition
The composition of the manuscript is not quite at the stage that would be ready for publication.
There are repetitive elements (e.g. Line 41 "...poly-unsaturated fatty acids, carotenoids and polyphenols,..." then Line 47 "...the omega-3 fatty acid eicosapentaenoic acid (EPA), polyphenols like (epi-) catechin and oxygenated carotenoids...".)
A section of the instructions is still in the manuscript (Line 177 "Results This section may be divided by subheadings. It should provide a concise and precise description of the experimental results, their interpretation as well as the experimental conclusions that can be drawn.")
Table 3 has two lines that are repeated (Fucoxanthin 10 and 1 ug/mL)
Some of the language used is not scholarly (eg "ingredients" "nowadays" "f.c. final concentration").
Some information is extraneous e.g. the date of cultivaton "from May to October 2017".
Although the meaning is understandable throughout the manuscript some phrasing could be improved eg "...can be grown sustainable" in the first sentence of the abstract should be "... can be grown sustainably."

Author Response

Dear Reviewer,

Thank you for your feedback and comments. Enclosed you will find detailed answers regarding your comments. Changes in the manuscript are highlighted using the "Track Changes" function in Microsoft Word.

1 MTT assays

The concentrations of FXN uses are fairly high and range from ~76 microM at 50 microgram/mL to the lowest concentration of ~152 nM at 0.1 microgram/mL.  Unsurprisingly there is a reduction in MTT activity at the higher doses in cell lines.  PBMC have a relative lesser reduction, but would probably be less active than cell lines.

-          As shown by Islam et al. (2013, see reference 36) concentrations of up to 100 µM fucoxanthin led to no significant decrease of cell viability in RAW264.7 cells. Our findings, however, show an effect at much lower concentrations, suggesting that small contaminations or that the source of the carotenoid might also have an effect.

Due to the generally lower metabolic activity of PBMCs, we used 3 x 105 cells per well and for RAW cells only 1 x 104 cells per well. This information was added to the text (line 108-109).

The use of the MTT assay here is asserted to be as a measure of metabolic activity, but this assay can also be used to infer the general health of the cells and can explain why functional activity is reduced.  In RAW cells cytokine gene expression was measured after treatment with FXN (Table 3) at a concentration that caused a 25% reduction in MTT activity (Table 2, 10ug/mL).  Given the caspase activity observed in HepG2 and HeLa cells (Fig 2) it is likely that FXN at these concentrations is causing cell death possibly by apoptosis.  This may account for the increased expression of TNF and IL1b expression at the 10ug/mL concentration, as dying cells releasing cellular content promote an inflammatory response if controlled apoptosis is not in effect.

-          We do believe that this comment is partly caused by a misunderstanding which we would like to explain. In RAW 264.7 cells only NO production was measured, the mRNA expression of IL-1β, IL-6 and TNFα was measured in PBMCs. In RAW 264.7 cells we saw no significant differences between the different groups, although a non-significant decrease in MTT activity was seen in the 10 µg/mL group.  A higher concentration (50 µg/mL) led to a significant decrease in NO production. However, we decided to exclude these data because this decrease is probably linked to the cytotoxic effect that was shown for this concentration in the MTT assay.
Since the MTT assay showed no effect of fucoxanthin on PBMCs we assume it is unlikely that used concentrations induce cell death. The mRNA expression of TNFα varied between different subjects which can also be seen in the high standard deviations. However, the increase was not significant for any concentration.

2 Inflammatory model

The stimulation of RAW cells with LPS in differing concentrations of FXN does not have a "no LPS stimulation" control.  This is necessary as the extract of FXN may activate cells of its own accord. Given the relatively lesser effect that FXN has on PBMC compared to RAW cells it is surprising that the authors did not use LPS-stimulated PBMC. By carefully collecting most of the supernatant after 24hr LPS stimulation for cytokine ELISAs, the PBMC could have been used for MTT assay.

-          Thank you for your helpful comment. A “no LPS stimulation control” was conducted for all assays and results were added to Table 2. Neither fucoxanthin nor β-carotene led to an increase in mRNA-expression leading to the assumption that no activation of cells was induced. As pointed out before PBMCs were used for evaluation of mRNA expression. This information was also included in Table 2.

3 Apoptosis
Figure 2 indicates that there may be degrees in the efficacy of the apoptosis-inducing effect of FXN.  The effect appears to be weaker in Caco-2 cells, and by comparing with HeLa and HepG2 cells the authors may better be able to dissect the mechanism by which FXN induces its effects.  This data is the most compelling but is not particularly novel given previous work in this area (PMID:28373414).  These assays are described as "proliferative assays" but could only be described as indicative of proliferation.  Proliferation is better measured by entry into S-phase (BrdU assay or thymidine incorporation) or cell fission (dye dilution in CFSE labelled cells) or inferring cell division by DNA content (propidium iodide staining).

-          Thank you for your comment. We have now rewritten the manuscript and described the assays as cytotoxic and apoptotic assays because no proliferative assays as described above were conducted.

4 Composition
The composition of the manuscript is not quite at the stage that would be ready for publication.
There are repetitive elements (e.g. Line 41 "...poly-unsaturated fatty acids, carotenoids and polyphenols,..." then Line 47 "...the omega-3 fatty acid eicosapentaenoic acid (EPA), polyphenols like (epi-) catechin and oxygenated carotenoids...".)

-          Thank you for your comment. We eliminated repetitive elements.

A section of the instructions is still in the manuscript (Line 177 "Results This section may be divided by subheadings. It should provide a concise and precise description of the experimental results, their interpretation as well as the experimental conclusions that can be drawn.")

-          Removed.

Table 3 has two lines that are repeated (Fucoxanthin 10 and 1 ug/mL)

-          Removed.

Some of the language used is not scholarly (eg "ingredients" "nowadays" "f.c. final concentration").
Some information is extraneous e.g. the date of cultivaton "from May to October 2017".
Although the meaning is understandable throughout the manuscript some phrasing could be improved eg "...can be grown sustainable" in the first sentence of the abstract should be "... can be grown sustainably."

-          We removed the extraneous information and used a more scholarly language.

Reviewer 2 Report

The manuscript “ Fucoxanthin derived from the pennate diatom Phaeodactylum tricornutum exerts antiproliferative and antioxidant activities in vitro” show interesting research issues. In the present manuscript authors have achieved his goal and contribute to the  knowledge of the fucoxanthin (from Phaeodactylum tricornutum) exerts antiproliferative and antioxidant activities. The research methods are acceptable, also includes statistical analysis. Selection of cell lines for tests very suitable.

Specific comments

L.99. Local ethics  committee - please add the decision number.

L. 131. dry weight or dry matter ?

L.178-180. Please remove.

2. Materials and Methods

Analyzes were repeated ?

L.181. 3. Results

The results should be interpreted based on significance. Please add significance for parameter changes. There is no need to give numerical data so often (they are in tables).

Generally the discussion is complete.

L. 307-309. Sentence does not follow from the research.

Author Response

Dear Reviewer,

Thank you very much for your positive feedback and your helpful comments. Enclosed you will find detailed answers regarding your comments

-          L.99. Local ethics  committee - please add the decision number.

The number was added, as well as the name of the local ethics committee.

-          L. 131. dry weight or dry matter ?

Thank you for your comment. We changed to dry matter.

-          L.178-180. Please remove.

Done.

-          Analyzes were repeated ?

Analyzes were repeated according to the n stated in figures and tables. For PBMCs each n represents one healthy volunteer. For cell cultures a multitude of subcultures was generated and each measurement was conducted with another subculture.

-          The results should be interpreted based on significance. Please add significance for parameter changes. There is no need to give numerical data so often (they are in tables).

Thank you for your comment. We added the information that p < 0.05 was used in ANOVAs as statistical significant differences. Additionally, some numerical data  that are also represented in tables were removed.

-          L. 307-309. Sentence does not follow from the research.

The sentence was changed to: “This leads to the conclusion that fucoxanthin or the whole microalgae biomass, including fucoxanthin in high amounts, could be considered in nutrition in order to ameliorate the effects of diseases linked to oxidative stress. Since the fucoxanthin used in this study was extracted from microalgae biomass, we assume that it now follows the research.

Reviewer 3 Report

Dear Editor,

The manuscript submitted by Ulrike Neumann et al. and entitled:” Fucoxanthin derived from the pennate diatom Phaeodactylum tricornutum exerts antiproliferative and antioxidant activities in vitro” is an interesting study which deals with extraction of bioactive diatom molecule. This manuscript can be accepted after minor revisions. Please to see my comments below.

Comments:

Authors should rephrase the title for a better understanding in the revised manuscript.

In material and method section, authors should give more details about extraction procedure.

Author Response

Dear Reviewer,

Thank you for your positive feedback and comments. Enclosed you will find detailed answers regarding your comments:

-          Authors should rephrase the title for a better understanding in the revised manuscript.

The title was rephrased “Fucoxanthin, a carotenoid derived from the diatom Phaeodactylum tricornutum, exerts antiproliferative and antioxidant activities in vitro” to give a better understanding.

-          In material and method section, authors should give more details about extraction procedure.

-          We added a more detailed description including an additional reference.

“The biomass was disrupted using stirred ball milling (PML-2, Bühler) and freeze-drying prior to fucoxanthin extraction. The cell disruption and extraction method was previously described in detail in Derwenskus et al. [17]. Briefly, Fucoxanthin was extracted from the disrupted biomass by pressurized liquid extraction (ASE 350, Thermo-Fisher) for a static extraction time of 20 min at 100 °C and 100 bar using adequate subcritical organic
solvents (described in [28]). Subsequently, the fucoxanthin was purified by multiple separation steps using filters (0,25 µm) consisting of polytetrafluoroethylene to a final purity of 99,2 % (w/w) (HPLC). It was compared to a commercial fucoxanthin standard (16337, Sigma-Aldrich) using UHPLC-MS.”

Round 2

Reviewer 1 Report

The manuscript is now at an acceptable standard but requires further fine checking for errors. 

Eg Line 231 "Hoewever, according to statistical analysis this change was not significantly (p = )."

Author Response

Dear Reviewer,

thank you for the second revision of our manuscript. We spell checked the manuscript and added the p-value in line 231.